# COMPRESSION BY THE SIGNS:
# DISTRIBUTED LEARNING IS A TWO-WAY STREET

**Jeremy Bernstein**[*]
California Institute of Technology
bernstein@caltech.edu

**Yu-Xiang Wang**
Amazon AI
yuxiangw@amazon.com

**Kamyar Azizzadenesheli**
University of California, Irvine
kazizzad@uci.edu

**Anima Anandkumar**
Amazon AI
anima@amazon.com

## ABSTRACT

Training large neural networks requires distributing learning over multiple workers. The rate limiting step is often in sending gradients from workers to parameter server *and back again*. We present SIGNSGD with majority vote: the first gradient compression scheme to achieve 1-bit compression of worker-server communication *in both directions* with *non-vacuous theoretical guarantees*. To achieve this, we build an extensive theory of sign-based optimisation, which is also relevant to understanding adaptive gradient methods like ADAM and RMSPROP. We prove that SIGNSGD can get the best of both worlds: compressed gradients *and* SGD-level convergence rate. SIGNSGD can exploit mismatches between $\ell_1$ and $\ell_2$ geometry: when noise and curvature are much sparser than the gradients, SIGNSGD is expected to converge at the same rate or faster than full-precision SGD. Measurements of the $\ell_1$ versus $\ell_2$ geometry of real networks support our theoretical claims, and we find that the momentum counterpart of SIGNSGD is able to match the accuracy and convergence speed of ADAM on deep Imagenet models.

## 1 INTRODUCTION

Training deep networks can be accelerated by distributing learning over multiple GPUs. Li et al. (2014) describe a popular framework where gradients from each GPU are sent up to a parameter server, aggregated and sent back down to the GPUs. The communication cost can be greatly reduced by compressing gradients. Here we propose and analyse a scheme for gradient compression where gradients are compressed *in both directions of transport* between parameter server and GPUs.

In our scheme, workers (i.e. GPUs) vote on the sign of the true gradient. The parameter server counts the votes and sends the majority decision back to each worker. Therefore conceptually the parameter server holds a referendum at every iteration to estimate the true sign of the gradient. Remarkably, under natural assumptions that are validated by experiment, we prove that majority vote as described converges at the same theoretical rate as full-precision distributed SGD.

The first step in our theoretical journey is to rigorously characterise the properties of sign-based methods for non-convex optimisation. We describe the geometry of a class of objective functions where SIGNSGD is expected to converge at equal or better rate than SGD. We expect these insights to be relevant to other adaptive gradient methods like RPROP, RMSPROP and ADAM.

We also extend our theoretical framework to the SIGNUM optimiser—which takes the sign of the momentum. Our theory suggests that momentum may be useful for controlling a tradeoff between bias and variance in the estimate of the stochastic gradient. On the practical side, we show that SIGNUM easily scales to large Imagenet models, and provided the learning rate and weight decay are tuned, all other hyperparameter settings—such as momentum, weight initialiser, learning rate schedules and data augmentation—may be lifted from an SGD implementation.

---

[*]Work carried out at Amazon AI, Palo Alto.

## 2 THEORY

We make assumptions that are fine-grained enough to encode heterogeneous curvature and sparsity, which SIGNSGD naturally exploits. *Our assumptions imply the standard* SGD *assumptions* with gradient Lipschitz constant $L := \|\vec{L}\|_\infty$ and total variance bound $\sigma^2 = \|\vec{\sigma}\|_2^2$.

**Assumption 1** (Lower bound). *For all $x$ and some constant $f^*$, we have objective value*

$$f(x) \geq f^*.$$

**Assumption 2** (Smooth). *Let $g(x)$ denote the gradient of the objective $f(.)$ evaluated at point $x$. And let $\delta$ be an upper bound on our learning rate. Then $\forall x, y$ satisfying $\|x - y\|_\infty \leq \delta$ we require that for some non-negative constant $\vec{L} := [L_1, ..., L_d]$*

$$\left| f(y) - \left[ f(x) + g(x)^T(y - x) \right] \right|$$
$$\leq \frac{1}{2} \sum_i L_i (y_i - x_i)^2.$$

**Assumption 3** (Variance bound). *Upon receiving query $x \in \mathbb{R}^d$, the stochastic gradient oracle gives us an* independent *unbiased estimate $\tilde{g}$ that has coordinate bounded variance:*

$$\mathbb{E}[\tilde{g}(x)] = g(x), \qquad \mathbb{E}\left[ (\tilde{g}(x)_i - g(x)_i)^2 \right] \leq \sigma_i^2$$

*for a vector of non-negative constants $\vec{\sigma} := [\sigma_1, .., \sigma_d]$.*

---

**Algorithm 1:** SIGNSGD

**Input:** learning rate $\delta$, current point $x_k$
$\tilde{g}_k \leftarrow \text{stochasticGradient}(x_k)$
$x_{k+1} \leftarrow x_k - \delta \, \text{sign}(\tilde{g}_k)$

---

**Algorithm 2:** SIGNSGD with majority vote

**Input:** learning rate $\delta$, current point $x_k$, # workers $M$ each with an independent gradient estimate $\tilde{g}_m(x_k)$
**on** server
    **pull** $\text{sign}(\tilde{g}_m)$ **from** each worker
    **push** $\text{sign}\left[ \sum_{m=1}^M \text{sign}(\tilde{g}_m) \right]$ **to** each worker
**on** each worker
    $x_{k+1} \leftarrow x_k - \delta \, \text{sign}\left[ \sum_{m=1}^M \text{sign}(\tilde{g}_m) \right]$

---

**Theorem 1** (Non-convex convergence rate of SIGNSGD). *Run algorithm 1 for $K$ iterations under Assumptions 1 to 3. Set the learning rate and mini-batch size (independently of step $k$) as*

$$\delta_k = \frac{1}{\sqrt{\|\vec{L}\|_1 K}} \qquad n_k = K$$

*Let $N$ be the cumulative number of stochastic gradient calls up to step $K$, i.e. $N = O(K^2)$. Then we have*

$$\mathbb{E}\left[ \min_{0 \leq k \leq K-1} \|g_k\|_1 \right]^2$$
$$\leq \frac{1}{\sqrt{N}} \left[ \sqrt{\|\vec{L}\|_1} \left( f_0 - f_* + \frac{1}{2} \right) + 2\|\vec{\sigma}\|_1 \right]^2$$

---

**Theorem 2** (Non-convex convergence rate of distributed SIGNSGD with majority vote). *Run algorithm 2 for $K$ iterations under Assumptions 1 to 3. Set the learning rate and mini-batch size for each worker (independently of step $k$) as*

$$\delta_k = \frac{1}{\sqrt{\|\vec{L}\|_1 K}} \qquad n_k = K$$

*Then (a) majority vote with $M$ workers converges at least as fast as SIGNSGD in Theorem 1.*

*And (b) further assuming that the noise in each component of the stochastic gradient is unimodal and symmetric about the mean (e.g. Gaussian), majority vote converges at unilaterally improved rate:*

$$\mathbb{E}\left[ \min_{0 \leq k \leq K-1} \|g_k\|_1 \right]^2$$
$$\leq \frac{1}{\sqrt{N}} \left[ \sqrt{\|\vec{L}\|_1} \left( f_0 - f_* + \frac{1}{2} \right) + \frac{2}{\sqrt{M}} \|\vec{\sigma}\|_1 \right]^2$$

*where $N$ is the cumulative number of stochastic gradient calls per worker up to step $K$.*

---

For proofs as well as a detailed discussion of all theoretical results, please see the full paper here: https://jeremybernste.in/projects/amazon/signum.pdf. The paper also contains a proof of the convergence rate for SIGNUM, which takes the sign of the momentum. The theorem suggests that momentum may be used to control a bias-variance tradeoff in stochastic gradient estimates.

## 3 EXPERIMENTS

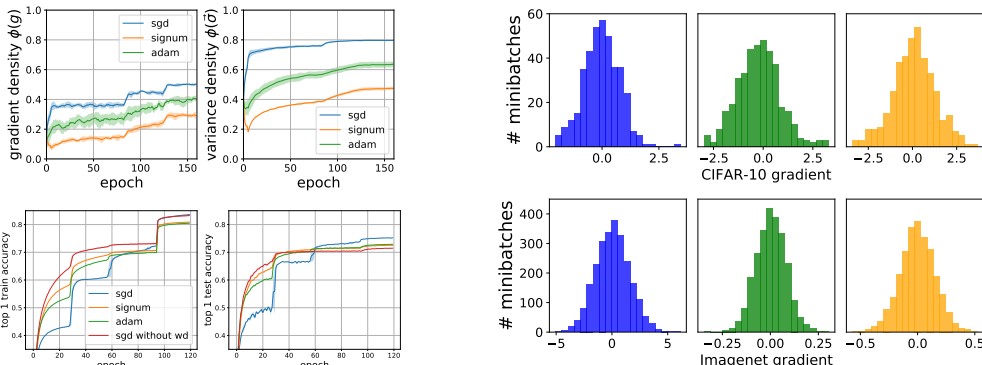

Figure 1: Top left: gradient and noise density during training of Resnet-20 on CIFAR-10, evaluated using Welford's algorithm (Welford, 1962; Knuth, 1997). $\phi(.) = 0$ is fully sparse and $\phi(.) = 1$ is fully dense. Since gradient and noise density are of the same order, our theory suggests that SIGNSGD should be competitive with SGD. Bottom left: indeed SIGNUM—the momentum version of SIGNSGD—is a competitive Imagenet optimiser. Right: stochastic gradient distributions at epoch 50 of training, for three non-cherry-picked weights. CIFAR-10 corresponds to training Resnet-20 with batch size 128. Imagenet corresponds to training Resnet-50 with batch-size 256. The stochasticity is approximately unimodal and symmetric in both cases.

## 4 DISCUSSION

In the theory section we derive the non-convex convergence rate of SIGNSGD and SIGNSGD with majority vote. There are two crucial aspects of the theory. First, the convergence rates naturally depend on the $\ell_1$-norm of gradients, curvature and stochasticity. Contrast this to SGD results where the naturally induced geometry is $\ell_2$. $\ell_1$ and $\ell_2$ norms can differ by as much as a factor of dimension, depending on the sparsity/density of typical vectors. This suggests that when the typical density of all relevant vectors is of the same order, then SIGNSGD should be naturally competitive with SGD, whilst also enjoying the benefits of gradient quantisation. Figure 1 suggests that gradient and noise densities are indeed of the same order for deep networks.

More heuristic gradient compression schemes like TERNGRAD (Wen et al., 2017) quantise gradients into three levels $\{0, \pm 1\}$. This can sometimes be desirable, and in practical settings we may wish to integrate ternary quantisation with our framework of majority vote. Our scheme should easily enable ternary quantisation—in both directions. This can be cast as "majority vote with abstention". The scheme is as follows: workers send their vote to the parameter server, unless they are very unsure about the sign of the true gradient in which case they send zero. The parameter-server counts the votes, and if quorum is not reached (i.e. too many workers disagreed or abstained) the parameter-server sends back zero. This extended algorithm should readily fit into our theory.

SIGNSGD and SIGNUM, like ADAM, are members of the family of adaptive gradient methods. In all our experiments we find that SIGNUM and ADAM get extremely similar performance, although both lose out to SGD by about 2% test accuracy on Imagenet. Wilson et al. (2017) also observed that ADAM tends to generalise slightly worse than SGD. It is still unclear whether this is due to the experimental baselines being biased towards models where SGD had previously been found to work well, or whether there is a deficiency in adaptive gradient methods like ADAM. Perhaps SIGNUM and ADAM could be generalising slightly worse because we don't know how to properly regularise such methods. One idea, suggested by our theory, is that SIGNSGD could be squashing down noise levels. There is some empirical evidence, for example by (Smith & Le, 2018), that a certain level of noise can be good for generalisation, biasing the optimiser towards wider valleys in the objective function. Perhaps, then, adding Gaussian noise to the SIGNUM update might help it generalise better. This can be achieved in a communication efficient manner in the distributed setting by sharing a random seed with each worker, and then generating the same noise on each worker. We leave this idea for future work.

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
