# OpenReview forum: "Compression by the signs: distributed learning is a two-way street"
_ICLR.cc/2018/Workshop — Accept_

### Official Review · AnonReviewer3 · 2018-03-06
**Nice paper**

**Rating:** 7
**Confidence:** 5

**Review:**

This paper discusses SignGD, an alternation of GD that uses only the sign of the gradient components. Under several assumptions, the authors provide guarantees that ensure convergence to a stationary point. They also compare their method to SGD for a deep learning task.

Theoretical part:
The theoretical part is nice and illustrates the convergence rate of SignGD to a stationary point.
Unfortunately, it seems like in theory SignGD  is slow compared to GD. Nevertheless, SignGD has a benefit in the distributed setting since we only have to communicate signs rather then full gradients.


Experimental part:
-The experiments are interesting and show that SignGD has comparable convergence to Adam and GD.


I recommend to accept the paper to ICLR workshop.

---

### Official Review · AnonReviewer1 · 2018-03-11
**A solid contribution, but needs more explanation**

**Rating:** 7
**Confidence:** 3

**Review:**


In this paper, the authors provide a series of new statistical algorithms which are particular suitable for distributed learning. The key difference lies between the proposed algorithm and the traditional SGD is that the update only relies on the sign of the stochastic gradient. Due to such update rule, the algorithm only requires 1-bit information about the gradient estimation, therefore, the communication cost will be largely reduced. The authors also provide the convergence guarantee of the algorithm, as well as the empirical comparison on the ResNet-20 on CIFAR-10, where the proposed algorithm achieves better results.

I vote for acceptance of this paper. My only concern is the explanation of the algorithm and the convergence theorem. Specifically, the intuition why only the sign of the gradient estimation is enough for the convergence of the algorithm is not clear. I understand the details of the technical proof of the convergence are omitted due to the space limitation. However, to convey a fully reasonable idea, at least the intuitation and the interpretation of the theorem should be added.

Secondly, as one of the major benefits of such algorithm is its efficiency in terms of the communication cost, it will be great if the empirical communication cost is compared with SGD and its variants, e.g., RMSprop and ADAM.

---

> ### Public Comment · ~Jeremy_Bernstein1 · 2018-03-21
> **Thanks for reviewing the work**
>
> Dear anonReviewer, thanks a lot for your feedback.
>
> The main observation about our theoretical bounds is that they depend on L1 norms of gradient, noise and stochasticity. Depending on how "well-distributed" these quantities are across the dimensions of the problem (i.e. ratio of L1 norm to L2 norm), signSGD can either be expected to perform better or worse than SGD. For Resnet-20 we show (Fig. 1, top-left) that noise and gradient are similarly "well-distributed", consistent with the empirical observation that signSGD converges at approx the same speed as SGD for deep nets.
>
> In Fig.1, bottom-left, we compare the algorithms on Imagenet.
>
> We provide all proofs and a much more detailed discussion of the intuition here: https://arxiv.org/abs/1802.04434
>
> Thanks again!

---

### Decision · Program_Chairs · 2018-03-20
**ICLR 2018 Workshop Acceptance Decision**

**Decision:**

Accept

**Comment:**

Congratulations, your paper was accepted to the ICLR workshop.